# The Effects of Deep Brain Stimulation on Balance in Parkinson’s Disease as Measured Using Posturography—A Narrative Review

**DOI:** 10.3390/brainsci15050535

**Published:** 2025-05-21

**Authors:** Bradley Lonergan, Barry M. Seemungal, Matteo Ciocca, Yen F. Tai

**Affiliations:** 1Department of Brain Sciences, Imperial College London, London W6 8RF, UK; bjl111@ic.ac.uk (B.L.); b.seemungal@imperial.ac.uk (B.M.S.);; 2Department of Neurology, Charing Cross Hospital, Imperial College Healthcare Trust (ICHT), London W2 1NY, UK

**Keywords:** Parkinson’s disease, posturography, deep brain stimulation, balance, postural instability

## Abstract

Background: Postural imbalance with falls affects 80% of patients with Parkinson’s disease (PD) at 10 years. Standard PD therapies (e.g., levodopa and/or deep brain stimulation—DBS) are poor at improving postural imbalance. Additionally, the mechanistic complexity of interpreting postural control is a major barrier to improving our understanding of treatment effects. In this paper, we review the effects of DBS on balance as measured using posturography. We also critically appraise the quantitative measures and analyses used in these studies. Methods: A literature search was performed independently by 2 researchers using the PUBMED database. Thirty-eight studies are included in this review, with DBS at the subthalamic nucleus (STN-) (*n* = 25), globus pallidus internus (GPi-) (*n* = 6), ventral intermediate nucleus (VIM)/thalamus (*n* = 2), and pedunculopontine nucleus (PPN) (*n* = 5). Results: STN- and GPi-DBS reduce static sway in PD and mitigate the increased sway from levodopa. STN-DBS impairs automatic responses to perturbations, whilst GPi-DBS has a more neutral effect. STN-DBS may promote protective strategies following external perturbations but does not improve adaptation. The evidence regarding the effects on gait initiation is less clear. Insufficient evidence exists to make conclusions regarding VIM- and PPN-DBS. Conclusions: STN- and GPi-DBS have differing effects on posturography, which suggests site-specific and possibly non-dopaminergic mechanisms. Posturography tests should be utilised to answer specific questions regarding the mechanisms of and effects on postural control following DBS. We recommend standardising posturography measures and test conditions by expert consensus and greater long-term data collection, utilising ongoing DBS registries.

## 1. Introduction

Parkinson’s disease (PD) is the second most common neurodegenerative condition globally, and modelling suggests that more than 25 million people will be living with PD by 2050 [1,2]. It is a disease which can manifest with a combination of motor (e.g., bradykinesia, tremor) and non-motor (e.g., depression, dysphagia) features. PD is diagnosed clinically, based on the classical motor features in the MDS diagnostic criteria, though imaging (e.g., dopamine transporter scans) can also support diagnosis [3]. There are multiple contributing factors to the pathogenesis of PD, including genetic and environmental, but the majority of cases are idiopathic. Inflammation and immunological changes, including microglial activation as described by McGeer, probably play a key role in PD initiation and propagation [4,5]. Novel research areas for pathogenesis include the role of nutrition and orexin, a neuropeptide linked to appetite [6,7]. Although PD can be grouped into tremor-dominant (TD) or postural instability gait disorder (PIGD) subtypes, individuals often have overlapping features or can switch from one subtype to another [8]. Postural imbalance with falls affects 80% of patients with PD 10 years following diagnosis [9]. This has major implications for people living with PD, as falls increase morbidity and mortality. The mainstay of PD treatment remains dopaminergic medications, though functional neurosurgery (e.g., deep brain stimulation [DBS] or lesioning surgery) is considered for advanced PD. These treatments have generally shown poor effects on postural instability, making it a topic of high clinical importance.

Many of the networks involved in postural control are pathologically affected by PD. The supraspinal pathways of postural control, in health and in PD, are displayed in Figure 1. Subthalamic nucleus (STN) activity is enhanced in PD due to loss of striatal dopamine from synucleinopathy, leading to excessive inhibition of the pedunculopontine nucleus (PPN) in the pons [10]. The PPN plays a key role in processing and integrating sensorimotor information and in maintaining the axial tone required for upright standing [11]. The degeneration of cholinergic neurons in the PPN in PD has been shown to correlate with falls in both post-mortem [12] and ante-mortem PET studies [13]. The PPN modulates the medullary reticular formation, which contains various non-dopaminergic neurons and is directly connected to central pattern generators in the spinal cord [14]. The cerebral cortex, particularly the frontal cortex, has also been implicated in postural responses via cortical-cerebellar and basal ganglia-cortical loops [15]. The cingulate sulcus visual area is responsible for integrating vestibular and optic inputs and has shown reduced activity on functional imaging in PD [16].

DBS at the STN and globus pallidus internus (GPi) are established treatments for fluctuations in the core motor features of PD (i.e., tremor, rigidity, bradykinesia). Lesioning functional neurosurgery for PD, including radio-frequency and MR-guided focused ultrasound (MRgFUS), are beyond the scope of this review. The mechanism by which DBS exerts its beneficial effects on these features remains debated. It has been suggested that the different core PD features arise from different oscillatory patterns in the basal-ganglia-thalamo-cortical network [17]. Stimulation causes neuronal activation in STN and GPi, demonstrated by electrochemical changes, leading to disruption of these abnormal networks [18]. Additionally, STN-DBS reduces levodopa-induced dyskinesia by reducing medication and/or disrupting the pathological pallidal outflow, whilst GPi-DBS has a more direct anti-dyskinetic effect via its output neurons [19,20,21].

However, the impact of DBS on postural instability and gait impairment is more complex. Falls and postural instability, especially when demonstrated in the ON state in the PIGD subtype, are usually contraindications to conventional high-frequency STN- and GPi-DBS, due to their propensity to worsen such features [22,23,24]. The mechanisms are likely multifactorial (e.g., disease progression) but could feasibly be affected by the spread of stimulation to other pathways known to be important in gait and posture, such as nigro-cuneiform nucleus pallido-PPN pathways [25]. PPN-DBS has also been trialled as a treatment for postural instability; despite some initial positive reports, subsequent results have been disappointing [26,27].

Although postural control and balance can be assessed clinically or using questionnaires, quantitative posturography gives richer details about the underlying mechanisms of postural impairments. Thus, we explore static and dynamic posturography following DBS in order to better understand the mechanisms by which DBS affects balance.

### 1.1. Static Posturography

Static posturography measures postural sway during quiet stance standing, without perturbation. This is usually performed on a force platform, as shown in Figure 2A, though wearable inertial measurement units (IMUs) can also be used. Strain gauges on the force platform measure distribution of pressure (or centre of pressure [CoP]) in medial–lateral (x) and anterior–posterior (y) directions as a proxy for sway. These forces can then be plotted against each other over time, as shown in Figure 2A. One benefit of IMUs is that the centre of mass (CoM) can be calculated more directly using accelerometer and gyroscope data, particularly if an IMU is worn on the trunk.

The key peripheral sensory inputs of balance (proprioceptive, visual, and vestibular) provide information about the environment which guides our orientation in space. To test individual sensory components, participants can be asked to stand on a soft surface (without proprioceptive feedback), with eyes closed (without visual feedback), or with both conditions to rely on vestibular function alone. Some computerised dynamic platforms (e.g., NeuroCom) can perform a sensory organisation test (SOT), which provides automated results from conditions with varying degrees of sensory input. The SOT test conditions are shown in Figure 2B. PD patients rely disproportionately on visual input for balance control; thus, they perform particularly poorly on static posturography with eyes closed [28]. The role of central processing is usually assumed during static posturography, as PPN function can only be measured indirectly, and its testing does not reflect the complexity of processing and integration [29,30].

As our posture strays beyond a certain angle from the vertical, it becomes progressively more difficult to bring ourselves back to the vertical without falling, unless we take a step. PD patients with excessive axial rigidity who fall may sway little during steady standing, whereas some patients with multiple falls manifest excessive sway [31,32]. This indicates a potentially problematic non-linearity in the relationship between sway and falls. Postural sway may be an overly simplistic measure and insufficient to extract meaningful physiological relevance to determine balance function. Static posturography has shown few differences between early PD and healthy controls, but sway increases as PD progresses, particularly in the lateral direction [33,34].

### 1.2. Dynamic Posturography

By contrast, dynamic posturography measures a person’s response to a physical perturbation. Test conditions, as shown in Figure 3, include external perturbations, voluntary leaning tasks and gait initiation. Participants’ responses are commonly measured as sway (using a force platform) or muscular activity (using electromyography [EMG]). These techniques have been described and reviewed elsewhere [35,36]. Dynamic posturography reflects real-life challenges more closely than static testing, so it may have greater utility in predicting actual falls, but it has been investigated less widely.

External perturbations can be delivered in different directions and via different mechanisms, but the most used is a moving surface delivering unidirectional random, unexpected translations [35]. External perturbations trigger ballistic preprogrammed movements that ideally involve stepping and/or configurational body alterations to stop us from falling. During formal dynamic testing, healthy individuals adapt their posture after detecting a perturbation and make predictive movements to reduce this imbalance [37,38]. There are various strategies that can be used, including head stabilisation, hip pivot, ankle pivot and whole-body rigidity strategies. Strategies change frequently with repeated perturbations [38]. Fallers are less likely to maintain their balance using an ankle pivot strategy [39]. PD patients tend to prefer an ankle pivoting strategy, though they may shift to a hip strategy when feeling more unbalanced [40,41]. PD patients also demonstrate a lack of adaptation to repeated perturbations [42]. This may be due to impaired utilisation of explicit cognitive strategies rather than implicit motor adaptation impairments related to basal ganglia dysfunction [43]. The relationship between postural strategy in response to external perturbations and overall balance or fall risk is not well established.

When someone decides to start walking, it triggers a set of anticipatory postural adjustments (APAs) before the first step is taken. The postural phase involves the body’s CoP moving away from then towards the stance leg (to maintain lateral stability) and posteriorly then anteriorly (to create forward movement) [44,45,46,47]. These movements are coordinated and scaled by the basal ganglia, thalamus, supplementary motor area and primary motor cortex [48,49,50]. Gait initiation APAs can be measured using CoP shifts from IMUs and lower limb motor responses from surface electromyography (EMG).

Leaning tasks assess speed and accuracy of leaning towards a target near one’s ‘limit of stability’. This is defined as the maximum distance that an individual’s CoM can be moved from central with feet in place without causing a step or fall [51]. An individual’s ‘limit of stability’ can be calculated using a computerised dynamic platform, though there are wide variations in methodology without an agreed consensus for measuring the ‘limit of stability’ [52].

Our working hypothesis is that there are inadequate biomarkers that delve fully into brain mechanisms of balance and linked clinical and home-based measures of falls in individual patients. To develop better approaches to measuring and monitoring imbalance (and its consequences) in PD, we review the use of quantitative measures of balance in PD via posturography, the analyses used and their relative utility following DBS. We specifically limit our analyses to those studies assessing the effects of DBS on postural control in PD and suggest how its utilisation can be improved to further our understanding of postural control.

## 2. Materials and Methods

A literature search was conducted on PUBMED, Scopus and Web of Science databases in March 2025, using the following MeSH terms: “deep brain stimulation”; “Posture”; “Postural balance”. Data were collected independently by two researchers (BL, MC) and then cross-checked to ensure included articles met inclusion and exclusion criteria and to avoid duplication.

The inclusion criteria for reviews were studies which measured instrumented static and dynamic posturography, with any metric, following DBS of any site in PD. Static tests include quiet standing sway and SOT. Dynamic tests include external perturbations (including automated systems such as Biodex), gait initiation and target acquisition with leaning. Studies were excluded for the following reasons, including varying combinations: duplication; only using clinical balance scales (e.g., Berg balance scale); animal studies; case studies; not including DBS patients; not measuring balance; dynamic posturography alone; non-PD DBS (e.g., essential tremor); non-DBS surgical interventions (e.g., surgical thalamotomy, MRgFUS); effects on camptocormia.

## 3. Results

Thirty-eight studies (15 static only; 16 dynamic only; 7 both static and dynamic) that met the inclusion and exclusion criteria were identified. Sixty-six percent of studies (25/38) explored the effects of STN-DBS alone in PD, compared to no DBS in PD or age-matched controls. Thirteen percent of studies (5/38) compared STN- and GPi-DBS in PD. All included studies are summarised in Appendix A within the Appendix A. All DBS cases were bilateral unless otherwise stated.

“Sway” is a non-specific umbrella term which includes various sway measures collected during posturography. There is no consensus on which sway measure most accurately reflects ‘balance’ or fall risk. The studies included in this review used sway path length, sway area and CoP displacement velocity most commonly. Appendix A describes and defines all the sway measures which were used during static posturography. Appendix A describes some additional measures which are used only during dynamic posturography. Where the term “sway” is used in this review, it may refer to any of these more specific measures (e.g., sway path length) from the original study. Many studies used both static and dynamic posturography, so the number of studies using each method combined exceeds the total number of studies.

The methodology relating to levodopa administration for the included studies was very variable (Appendix A). Differences include whether testing occurred in medication-ON/-OFF/both; time for medication withdrawal classified as medication-OFF; whether medication-ON meant supratherapeutic or optimised dosing; and whether dyskinesia severity was described or excluded.

The results are arranged with DBS target locations as subheadings; subsequent findings are ordered by describing the results of studies which used (a) static posturography and (b) dynamic posturography. Levodopa effects with DBS are explored within the relevant paragraphs.

### 3.1. Subthalamic Nucleus (STN)

Twenty-five studies investigated the effects of bilateral STN stimulation alone on static (*n* = 15) and dynamic (*n* = 14) posturography in PD (Appendix A).

Some studies showed that STN-DBS reduced quiet stance sway as measured by sway area [53], mediolateral sway [54], CoP velocity [55] or a combination of the above [56,57,58]. Other studies showed no significant effect of STN-DBS on static posturography, including a comparison of low- and high-frequency stimulation [59,60,61,62]. Szlufik et al. found non-significant reductions in sway 9 months after DBS, followed by significant sway increases after another 9 months [63].

Three studies reported significant findings during static posturography with varying degrees of sensory input (e.g., eyes closed, unstable platform) [55,56,64]. Reduced sway [56] and increased sway [64] were both reported during quiet standing with eyes closed, replicating the heterogeneous results in the literature [54,65]. Shivitz et al. split participants into those with AP sway that was normal (i.e., >5th percentile on the spectrum of controls) and abnormal (i.e., <5th percentile on the spectrum of controls), presented as equilibrium scores, in six conditions with varying degrees of sensory input [55]. In those with increased sway, STN-DBS reduced sway when they were in vestibular-dependent conditions; dopaminergic medication had no effect. Additionally, there were significant reductions in sway in more challenging conditions (e.g., eyes closed, unstable platform) after DBS with stimulation off. STN-DBS had no effect in those with normal sway.

Patel et al. were the only group to measure sway using ultrasound 3D motion capture and motion markers at anatomical landmarks (e.g., hip) instead of using a force platform. This measures the spectral power of movement at different frequencies during quiet standing. STN-DBS reduced sway >4 Hz laterally at head and shoulder level; there were no changes at the knee and hip [42,66,67]. There was little change at lower frequencies (0–4 Hz) [66]. Given that PD rest tremor occurs at 4–7 Hz, the authors hypothesised that the reduced sway was due to reduced tremor post-DBS.

Regarding dynamic testing, six studies have shown varying effects on the initiation of voluntary walking. STN-DBS increased APA amplitude prior to gait initiation, which improved foot lift, but had no effect on the speed of APA onset [44]. Two others, including a comparison of low and high frequency, found no significant impact on APAs but did improve gait parameters (e.g., step length) [62,68]. One group found that STN-DBS improved standard distance, as part of a principal component analysis, particularly when combined with levodopa. However, the sway improvements waned after 7 years due to PD progression [69,70]. Another principal component analysis of gait initiation measures suggests that directional DBS may be superior to traditional ring stimulation, whilst both are better than no DBS [71]. Directing stimulation towards the central STN and more central lead positioning had better gait outcomes than stimulating the posterior STN. The benefits of central stimulation compared to ring stimulation were less clear.

Five studies looked at response to balance perturbations after STN-DBS, including three studies which looked at body segment coordination and postural strategies [37,42,67]. One found increased coupling between body segments with STN-DBS-ON, suggesting an ankle strategy, which was greater with eyes open than eyes closed [42]. This comes at the cost of energy and flexibility but is probably helpful for maintaining balance. Additionally, visual input helped to stabilise posture more with STN-DBS-ON than DBS-OFF, suggesting DBS may help visual processing [67]. However, STN-DBS did not help adaptation, regardless of sensory input [67]. Following repeated calf muscle vibration, PD patients displayed greater flexion of the head, shoulder and knee than controls [37]. This was partially resolved with STN-DBS-ON. The other two studies delivered external perturbations on a force platform [72,73]. Leodori et al. used the Biodex system, which generates composite balance measures from participant’s responses (e.g., Stability Index, Risk of Falls). Bilateral STN stimulation and levodopa combined showed the best impact on these measures, compared to various unilateral and levodopa combinations. Despite a slight trend towards improvement, there were few significant differences between medication alone and combination with STN-DBS on muscle response latency and gastrocnemium/tibialis anterior co-contraction ratio following perturbations [73].

Three trials investigated the effects of STN-DBS on leaning tasks close to an individual’s ‘limit of stability’. Higher STN-DBS amplitudes increased leaning velocity, but velocity was slower at lower amplitudes than off stimulation [74]. This suggests a threshold effect for amplitude on leaning velocity. Increased success in target acquisition has also been reported post-operatively, with subsequent improvements in clinical balance scales (e.g., UPDRS) at 6 and 12 months post-operatively [75,76].

### 3.2. Globus Pallidus Internus (GPi)

Only one study investigated the effects of GPi-DBS alone on static and dynamic posturography (Appendix A). Johnson et al. found that GPi-DBS non-significantly reduced sway area but had no impact on sway PL or sway velocity [77]. Similar findings with GPi-DBS are reported elsewhere; sway area was significantly reduced, and displacement distance returned to age-matched healthy control levels [78]. Dynamically, GPi-DBS improved accuracy of leaning during a target acquisition task, with less effect on the time taken to start the task [77]. Levodopa and GPi-DBS had opposite effects on leaning; levodopa reduced the time taken to start moving, and GPi-DBS improved the accuracy of movement [77]. The levodopa effects may have been due to mild dyskinesia, as severe dyskinesia was excluded, and/or alternative unknown mechanisms.

### 3.3. GPi vs. STN

Five studies compared GPi- with STN-DBS; two of these used static posturography [76,77], and three used dynamic testing (Appendix A) [49,79,80]. Brandmeier et al. found no significant difference in sway index in those with or without DBS (GPi and STN) and no differences between GPi and STN groups [79]. GPi- and STN-DBS both seem to counteract the increase in sway seen after taking levodopa to some degree [53,58,62,72,77,78], though there are some conflicting results [57]. STN-DBS has a greater effect than GPi-DBS in counteracting these levodopa effects [77,78].

STN-DBS seems to worsen responses to external perturbations, whilst GPi-DBS has no effect. One study showed initial improvements in the stability of participants’ automatic postural response (APR) to external perturbations, whilst levodopa had no effect [81]. However, this improvement in perturbation response waned in the STN-DBS group, such that APR stability was worse at 6 months than it was at baseline [81]. STN-DBS also led to more falls, likely by prolonging the in-place preparation phase and delaying stepping [80]. GPi-DBS had no effect on falls, stability of response to perturbations or stepping response 6 months after DBS [80,81].

A study comparing gait initiation following STN- and GPi-DBS showed few differences between the two targets. Preparatory APAs were worse (smaller size and longer duration) after both STN- and GPi-DBS compared to pre-surgery and less responsive to the positive effects of levodopa post-DBS [49]. By contrast, actual step execution was largely unchanged by DBS at both sites, suggesting different mechanisms for step preparation and step execution [49].

### 3.4. Ventral Intermediate (VIM) Nucleus and Thalamic Tracts

One study investigated the effects of VIM-DBS on static and dynamic posturography in PD (Appendix A). Ondo et al. investigated bilateral VIM stimulation in Essential Tremor (ET) (13 patients) and PD (8 patients, all TD) [82]. In PD, sway improved significantly in SOT condition 4 (eyes open; sway-referenced support) and non-significantly in SOT condition 1 (eyes open; fixed support) after VIM-DBS. PD patients showed no adaptation to perturbations with or without VIM-DBS and greater variability of response to ET patients. Overall, they reported balance to be similar in both the ET and PD groups, though they also noted the high severity of the ET cohort.

Additionally, experimental stimulation of the Fields of Forel thalamic tract reduced sway and falls and improved clinical balance scale (e.g., Berg balance scale) scores in PD patients with levodopa-unresponsive gait disturbance [83].

### 3.5. Pedunculopontine Nucleus (PPN)

Five studies investigated the impact of PPN stimulation on static (*n* = 2) and dynamic (*n* = 3) posturography (Appendix A). Some studies investigated specific PD subgroups: levodopa-unresponsive freezing of gait (*n* = 2) and severe PIGD (*n* = 1).

Mazzone et al. found that PPN stimulation reduced sway path length with eyes open compared with no stimulation (self-control) in eight PD patients with severe PIGD. There were also other non-significant reductions in sway with eyes open and eyes closed [84]. PPN-DBS also reduces double stance duration and increases first step length/velocity and the size of APAs prior to the first step [85]. There seemed to be some additive effects from PPN-DBS with levodopa, compared to either treatment in isolation, including increased first step velocity [85].

Yousif et al. compared the effects of STN-ON/PPN-ON stimulation with STN-ON/PPN-OFF stimulation [86]. STN-ON/PPN-ON stimulation was associated with increased sway with EC compared to STN-ON/PPN-OFF, though levodopa was not controlled for. In contrast to most studies, the authors suggest that increased sway with EC may improve balance.

Bourilhon et al. performed two studies comparing PPN-DBS with cuneiform nucleus (CN)-DBS. At 2 months, step length and step velocity were significantly higher with CN-DBS than with sham-DBS, which were significantly higher than with PPN-DBS [87]. At 1 year, there were no significant differences compared to pre-operative testing. At 2 years, both locations led to increased double stance duration compared to pre-operatively [88]. In Medication-ON, PPN-DBS had significantly greater cadence, walking velocity and step length, and lower double stance duration and turn duration compared to CN-DBS [88]. There were no significant differences in Medication-OFF. APA duration and displacements do not appear to have significantly changed throughout.

## 4. Discussion

This review includes studies using quantitative measures of balance in PD via posturography following DBS (Appendix A). The different effects of DBS, particularly at the STN and GPi, and levodopa on static and dynamic posturography suggest that stimulation has effects beyond dopaminergic pathways. Some effects appear to be synergistic, (e.g., improving gait initiation) [69], whereas other effects appear contradictory (e.g., static sway) [54,58,77,78]. More recent studies have used larger group sizes; this is probably aided by the increased availability of wearable sensors which, whilst expensive, make it easier to collect patient data in clinical settings [62,73].

The greatest limitation of the available evidence is the high degree of heterogeneity across studies, because it prevents comparison of their results and leads to small sample sizes which lack the power to make definitive conclusions. Small sample sizes are compounded by the high resource cost of DBS and its availability being limited to tertiary centres. Sources of heterogeneity include test conditions (e.g., sensory inputs during static testing and perturbation type during dynamic testing), DBS anatomical site (e.g., PPN, STN), outcome (sway) measures (e.g., sway path length, sway area and CoP velocity) and control groups (e.g., absent, age-matched or disease-matched). Additionally, the studies included rarely differentiate between the PD subtypes (TD vs. PIGD); thus, it is difficult to comment on the differential effects of DBS on each subtype. The introduction of directional DBS will add additional complexity to analysing posturography post-DBS, but also greater promise of individualised programming for PwP, including discovering settings which may avoid postural instability as a DBS side effect.

Sway vector has been suggested as a more reliable and reproducible sway measure, as it can be successfully measured irrespective of trial length and sampling frequency [89]. Although it has been validated in PD, it is yet to make it from research to clinical settings. The International Society for Posture and Gait Research (ISPGR) and other groups have been unable to standardise the methodology of posturography, though their work is ongoing [89,90,91,92]. Only three studies collected data beyond 12 months; thus, we know little about the long-term effects of DBS on postural control and how this compares to disease progression without DBS [63,70,79]. DBS effects on sway may wane over time, so it is important that longer-term data are collected [63]. One solution would be to utilise DBS registries to collect more data on postural control following DBS; for example, a subset of registry participants could be invited to complete posturography.

### 4.1. Effects of DBS on Static Posturography

STN- and GPi-DBS reduce static sway; further work is needed to improve our understanding of the effects of VIM- and PPN-DBS. STN-DBS may reduce sway more during vestibular-dependent conditions in those with high sway at baseline [55]. This hints at greater complexity and individualised results than are currently accepted in the literature. The most common assumption about static posturography is that increased sway represents worse postural control and a higher risk of falling. This is likely an oversimplification; the role of sway may vary depending on context and neurological impairments. One alternative theory is that greater sway leads to greater activation of peripheral mechanoreceptors, providing greater sensory feedback from the peripheries and paradoxically improving balance [64,86]. The impacts of sensorimotor processing and integration on sway are usually assumed, rather than directly tested. Only one included study also formally tested peripheral sensory processing; PPN-DBS was shown to improve vestibular perceptual thresholds [86]. Neurophysiological techniques, such as the pre-pulse inhibition, should be used simultaneously to assess the impact of DBS on sensorimotor integration [93]. We predict that the mechanisms for DBS effects, which are currently poorly understood, on sway are different according to DBS targets.

The STN and GPi are both connected to the PPN via GABAergic indirect and direct basal ganglia pathways, respectively [94]. Differences between the make-up of the STN and GPi, such as the relative density of axons to cell bodies and response of single units to stimulation, the strength of connectivity to the PPN and the density of vestibular-connected neurons, could all feasibly lead to variable responses [20]. Stimulation of non-dopaminergic (e.g., cholinergic) networks, via the PPN or superior colliculus, may improve sensorimotor integration and/or postural control [30,77,78,95]. We hypothesise that DBS improves PPN feedback to the basal ganglia, which helps to reduce sway by improving postural control and tone. There are also other dopaminergic and cholinergic improvements via basal ganglia outputs, which help to improve motor features of PD such as rigidity. Given that GPi is downstream of STN, uninterrupted STN overactivity could theoretically make GPi-DBS less effective at improving postural control, but this is not supported by the available evidence. Figure 4 demonstrates how DBS at different sites may affect the supraspinal postural control network, according to our hypothesis. The wider network effects of GPi- compared to STN-DBS are poorly understood; however, the advent of MRI-compatible stimulators makes this a more achievable target for the future.

As well as potential direct DBS effects, there are indirect effects which may help to reduce sway, such as levodopa dosage reduction. Most of the included studies show that DBS (decreased sway) at least partially compensates for the effects of levodopa (increased sway) on postural control. Levodopa may increase sway by reducing axial muscle tone or by increasing levodopa-induced dyskinesia. DBS effects often mean the levodopa dose can be reduced, leading to reduced dyskinesia and reduced motor fluctuations [96]. Few studies in this review reported dyskinesia incidence, but these studies showed unchanged dyskinesia incidence post-DBS [57], low dyskinesia incidence from baseline [78] or excluded patients with severe dyskinesia [58]. GPi-DBS usually leads to less reduction in levodopa dose, so greater sway would be expected with GPi-DBS if the effect was solely affected by levodopa [49]. Levodopa increases sway less with STN- than GPi-DBS; these differential levodopa effects by DBS site need further exploration [77,78].

Reduction in tremor may explain reduction in sway across different DBS sites, even if tremor reduction is achieved via different pathways. For example, lateral movements at the head and neck in the PD rest tremor band (4–7 Hz) were reduced with STN-DBS [67]. The only study of VIM-DBS showed significant tremor improvements with mild increases in sway in some sensory conditions [82]. More studies are needed to explore whether VIM-DBS effects on tremor could simultaneously reduce sway.

Some studies also showed that sway is reduced with STN-DBS-OFF compared to pre-DBS insertion. This may be due to an acute microlesion effect, chronic STN stimulation effects which persist when the device is off, or a reduction in levodopa dose [55,70]. Most studies tested DBS-OFF at least 30 min after stimulation had been switched off, but there was some variety. The washout period for GPi-DBS may be longer than for STN-DBS, which affects their comparability [97]. There should be standardisation of DBS wash-out times across studies to improve consistency and comparability, as well as further investigation of the effects of chronic stimulation.

It is difficult to draw strong conclusions about the impact of PPN-DBS on posturography, given the lack of available data. Current results show that PPN-DBS reduces sway, particularly with levodopa, though effects also seem to wane over time [84,85,88]. Combined PPN-STN-DBS increased sway with EC; this suggests that STN- and PPN-DBS have different mechanistic effects on balance [86]. It is possible that increased sway improves balance control when visual input is lacking, as it provides greater sensory input to the system, particularly if sensory processing capacity has increased.

### 4.2. Effects of DBS on Dynamic Posturography

Dynamic posturography is an umbrella term for different techniques, which probably test different aspects of balance. It is unclear whether dynamic testing is more representative of postural control and/or real-life fall risk than static posturography. Whilst external perturbations directly test postural reflexes, leaning tasks probably test a wider range of factors (e.g., bradykinesia and postural/movement control). The postural adjustments that precede gait initiation probably form a different motor programme to the automatic adjustments that occur following perturbations. To improve the utility of these tests, studies should focus on the mechanisms of balance that are being tested and how the results can be applied to fall risk in real life.

GPi-DBS seems to have a neutral effect on response to external perturbations, whereas STN-DBS may have a negative impact. Although some studies showed non-significant trends towards improvements in muscle latency and automatic postural responses, balance and falls appeared to be worse 6 months after STN-DBS [42,67,72,73,81]. If this finding is replicated in future studies and consistent across other aspects of balance testing (e.g., adaptation, strategy), GPi-DBS may be preferable in individuals that have worse postural instability during pre-DBS testing. The impact on gait initiation is less clear, with conflicting results and less available evidence for GPi-DBS [44,62,68]. There may be differences between how DBS at both sites affects preparatory postural movements (negative) and actual step execution (neutral) [49]. Where balance impairments are seen post-DBS, experimenting with directional stimulation (e.g., towards the central STN) may help to mitigate these effects [71].

STN-DBS partially improves body alignment after a perturbation and increases the stabilising effect of vision, suggesting improved visual input processing [37,67]. STN-DBS increased body segment coupling and promoted an ankle strategy, both of which may be helpful in preventing falls [66]. Optimal strategy may vary according to the individual and the situation; further work is needed before particular strategies can be encouraged. STN-DBS had no effect on adaptation to repeated perturbations [67]. Rehabilitation which focuses on improving cognitive aspects and drawing attention to balance control may be of benefit [43].

STN-DBS beyond a threshold amplitude leads to a sustained improvement in leaning velocity [74,75,76]. GPi-DBS has been shown to improve leaning accuracy but not speed of leaning initiation [77]. The ecological benefits (or detriments) of these effects on voluntary leaning are unclear.

Although PPN-DBS may improve gait dynamics, this appears to wane over time from the limited evidence available [85,88]. The effects of VIM-DBS on dynamic balance and thalamic tract DBS, as an experimental treatment, have also been underexplored.

There are limitations to this review. Firstly, there may be an element of publication bias; studies with less significant results may not have been published, so real-world DBS effects may be different from those which are published. Secondly, methodological heterogeneity makes it difficult to make direct comparisons between included studies or to perform higher-level analysis (e.g., meta-analysis). Given the tertiary and high-cost nature of DBS, each study includes small numbers of participants. Finally, this review only includes instrumented posturography, rather than clinical scales (e.g., BESTest), which may also accurately measure postural control.

## 5. Conclusions

Postural instability and falls have a major impact on morbidity and mortality for PD patients and respond poorly to currently available PD treatments. To develop better treatments and optimise current treatments, such as DBS, we need to improve our understanding of postural control mechanisms and how postural control could be modulated. The available studies have limited overall utility due to their heterogenous methodology and our limited understanding of how their results should be interpreted. Posturography tests should be used to address specific questions regarding individual elements of postural control, rather than making general conclusions about balance.

Static posturography provides useful information on how balance-related sensory inputs (e.g., vision, proprioception, vestibular) affect static balance in an individual. STN- and GPi-DBS seem to reduce sway; this may be caused by any combination of reduced tremor, reduced dyskinesia, improved sensorimotor processing or unidentified factors and likely vary between anatomical sites. There is a lack of evidence to make conclusions about the effects of VIM- and PPN-DBS. The effects of DBS likely vary according to individual or contextual factors. For example, those with high sway at baseline seem to reduce sway more after DBS, particularly when visual input is lacking [51]. By contrast, dynamic tests should be used to demonstrate whether automatic sensorimotor loops are functioning adequately. This may be mediated by the PPN, given its role in sensorimotor processing, but the mechanisms are incompletely understood.

As well as supporting mechanistic research on postural control in PD, we make several practical recommendations for groups conducting posturography following DBS. Firstly, we support the ISPGR’s efforts to standardise static and dynamic posturography methodology. Rather than curbing innovation, this should be seen as a ‘minimum’ standard which is routinely collected when using a particular method. An international guideline which recommends a ubiquitous sway measure (e.g., sway vector) and testing conditions (e.g., perturbation speeds) would help provide direction and build a stronger evidence base in the future [98]. Secondly, given the success of international DBS registries, we suggest that a subset of participants be invited for posturography testing post-DBS. This would build a greater base of long-term posturography data post-DBS and improve sample sizes. Additionally, future studies should explore the impact of recent technological advances in DBS, such as directional and adaptive stimulation, and less common DBS programmes, such as low-frequency stimulation, on posturography. Finally, future studies should attempt to quantify the effects of DBS on different sensory inputs and their processing, as they are central to postural control. For example, only one included study measured vestibular perceptual thresholds [86]. DBS could feasibly affect sensorimotor processing, but this is an unexplored area.

## Figures and Tables

**Figure 1 brainsci-15-00535-f001:**
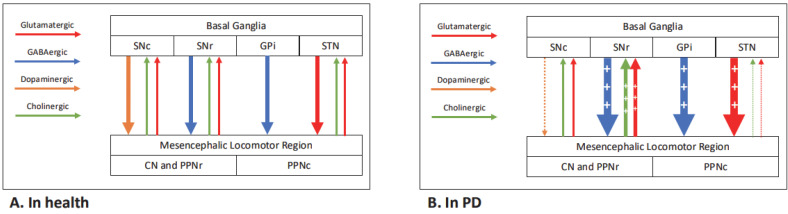
Dopaminergic and non-dopaminergic supraspinal pathways of postural control in health (**A**) and in PD (**B**). ‘+’ denotes overactivity, and ‘dashed line’ denotes underactivity with respect to normal physiology. The number of ‘+’ and ‘dashes’ relates to the degree of over- and underactivity. CN—cuneiform nucleus, GPi—globus pallidus internus, PPNc/r—pedunculopontine nucleus caudal/rostral, SNc/r—substantia nigra pars compacta/reticulata, STN—subthalamic nucleus.

**Figure 2 brainsci-15-00535-f002:**
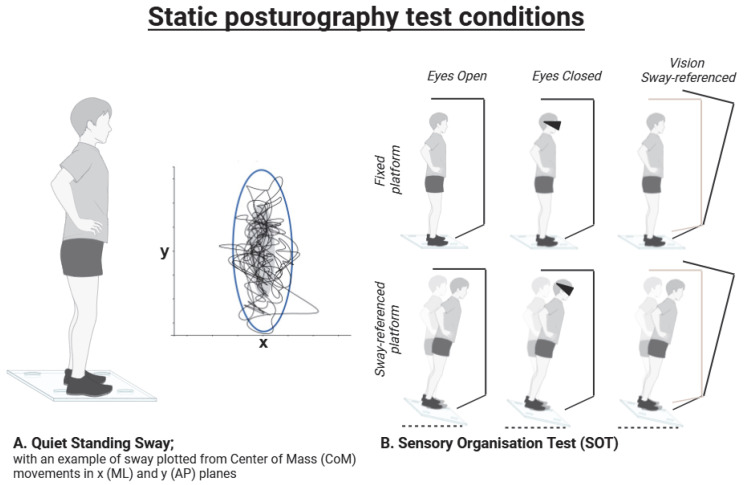
Common static posturography test conditions. Quiet standing sway is usually performed on a force platform, though IMUs can also be used; the graphic shows an example of CoM sway data generated from IMUs, expressed as x vs. y (**A**); Sensory organisation test (SOT) conditions, which measures sway during combinations of visual, somatosensory and sway-referenced conditions (**B**). AP—anteroposterior; ML—mediolateral. Created in BioRender. Lonergan, B. (2025). URL: https://BioRender.com/20nvui6 (accessed on 21 March 2025).

**Figure 3 brainsci-15-00535-f003:**
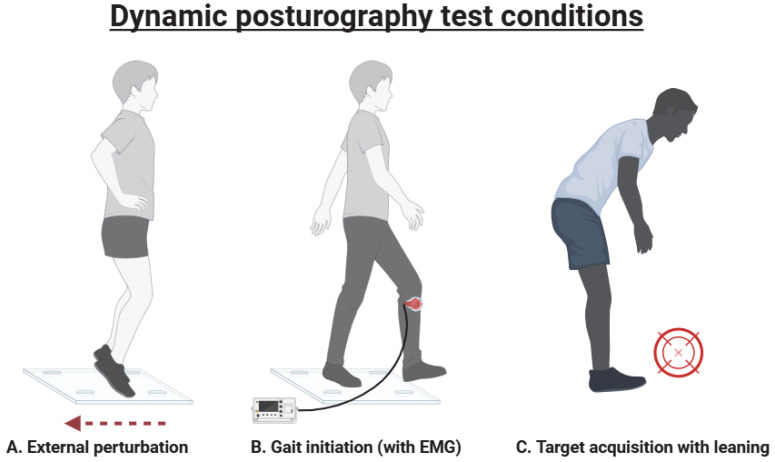
Common dynamic posturography test conditions include external perturbations (**A**), gait initiation (**B**) and target acquisition with leaning (**C**). Created in BioRender. Lonergan, B. (2025). URL: https://BioRender.com/dw6gsg6 (accessed on 21 March 2025).

**Figure 4 brainsci-15-00535-f004:**
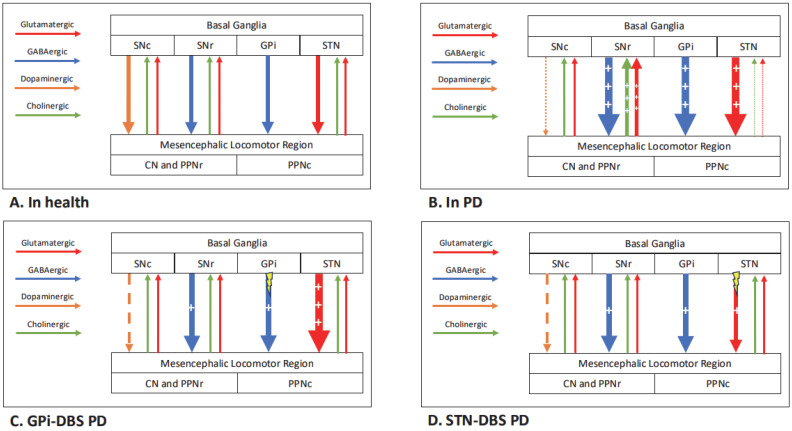
Dopaminergic and non-dopaminergic supraspinal pathways of postural control in health (**A**) and in PD (**B**), with proposed mechanisms for GPi- (**C**) and STN-DBS (**D**). ‘+’ denotes overactivity, and ‘dashed line’ denotes underactivity with respect to normal physiology. The number of ‘+’ and ‘dashes’ relates to the degree of over- and underactivity. CN—cuneiform nucleus, GPi—globus pallidus internus, PPNc/r—pedunculopontine nucleus caudal/rostral, SNc/r—substantia nigra pars compacta/reticulata, STN—subthalamic nucleus.

## Data Availability

No new data were created or analysed in this study. Data sharing is not applicable to this article.

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
