# Peer review of "The Effects of Deep Brain Stimulation on Balance in Parkinson’s Disease as Measured Using Posturography—A Narrative Review"

_brainsci, 2025, doi:10.3390/brainsci15050535_

Round 1

Reviewer 1 Report (Previous Reviewer 1)

Comments and Suggestions for Authors

The manuscript was revised when compared to the previous version from March 2025. I would suggest acknowledging the McGeer theory regarding the pathophysiology of Parkinson's Diseases. Apart from that i believe that this manuscript has merits of being published in the journal.

Author Response

Comment 1: I would suggest acknowledging the McGeer theory regarding the pathophysiology of Parkinson's Diseases.

Response 1: Thank you for your feedback and support in favour of publication. We have expanded the Background section relevant to pathophysiology to include the role of inflammation and to make reference to McGeer's theory (see lines 44-46, highlighted for clarity). We have also added new references (ref 4 & 5, also highlighted).

Reviewer 2 Report (Previous Reviewer 2)

Comments and Suggestions for Authors

The revised paper is generally well-written with clear structure and logical flow. However, there are minor issues affecting readability:

  1. Some sentences are overly complex (e.g., "Static posturography is defined as techniques that 'measure quiet standing...without any physical body perturbation' [27]."). Breaking these into shorter sentences would improve clarity.

  2. The phrase "Our working hypothesis is that..." is duplicated verbatim in the Introduction.

  3. For example, "This effect waned..." (Page 8) lacks a clear referent for "this."

  4. Terms like "sway path length" and "sway area" are used interchangeably without explicit definitions in some sections.

  5. References to figures (e.g., Figure 1 and 4) lack sufficient explanatory context, leaving readers to infer connections.

Overall, the language and content is academically appropriate, but minor revisions for conciseness and precision would enhance readability. The authors could consider language editing in their revision.

Author Response

Thank you for your helpful comments. We have made some changes to improve clarity and highlighted these changes to make them easy for you to identify.

Comment 1: Some sentences are overly complex (e.g., "Static posturography is defined as techniques that 'measure quiet standing...without any physical body perturbation' [27]."). Breaking these into shorter sentences would improve clarity.

Response 1: Thanks for identifying this. We have changed the start to sections 1.1 (lines 108-109) and 1.2 (line 148-153) to improve clarity.

Comment 2: The phrase "Our working hypothesis is that..." is duplicated verbatim in the Introduction.

Response 2: This duplication was an error and has been removed.

Comment 3: For example, "This effect waned..." (Page 8) lacks a clear referent for "this."

Response 3: We have tried to change both sentences that begin with 'This effect waned...' (lines 276 and 329) to try to improve clarity.

Comment 4: Terms like "sway path length" and "sway area" are used interchangeably without explicit definitions in some sections.

Response 4: Thank you for this comment. The lack of clarity in our review reflects the heterogeneity within the papers included in our review. We have added a few sentences (lines 225-232) to better describe how the term 'sway' is used in our review and the different ways in which 'sway' is measured throughout the literature.

Comment 5: References to figures (e.g., Figure 1 and 4) lack sufficient explanatory context, leaving readers to infer connections.

Response 5: We have changed all mentions of figures so that they are full sentences and fully explained, rather than just in brackets. We hope that this has improved their clarity.

This manuscript is a resubmission of an earlier submission. The following is a list of the peer review reports and author responses from that submission.

Round 1

Reviewer 1 Report

Comments and Suggestions for Authors

Lonergan et al provide a review on the significance of deep brain stimulation on balance in Parkinson’s Disease. I have the following comments regarding this work:

  1. In the introduction it would be valuable to provide initial information on the disease discussed in the work e.g. epidemiology, pathophysiology, subtypes - Ref. Role of orexin in pathogenesis of neurodegenerative parkinsonisms. Neurol Neurochir Pol. 2023;57(4):335-343. doi:10.5603/PJNNS.a2023.0044 / The epidemiological risk and prevention and interventions in Parkinson's disease: from a nutrition-based perspective. J Nutr. . doi:10.1016/j.tjnut.2025.01.028 / MDS clinical diagnostic criteria for Parkinson's disease. Mov Disord. 2015;30(12):1591-1601. doi:10.1002/mds.26424
  2. The possible significance of DBS in the context of balance should be discussed in the context of subtypes of PD - tremor-dominant/ postural instability 
  3. The information concerning used pharmacological treatment should be discussed more extensively
  4. The discussed feasibility of DBS should be discussed in the context of other stereotactic methods e.g. Gamma Knife and Focus Ultrasound.
  5. The description of methodological limitations is insufficient.

Author Response

Comment 1: In the introduction it would be valuable to provide initial information on the disease discussed in the work e.g. epidemiology, pathophysiology, subtypes - Ref. Role of orexin in pathogenesis of neurodegenerative parkinsonisms. Neurol Neurochir Pol. 2023;57(4):335-343. doi:10.5603/PJNNS.a2023.0044 / The epidemiological risk and prevention and interventions in Parkinson's disease: from a nutrition-based perspective. J Nutr. . doi:10.1016/j.tjnut.2025.01.028 / MDS clinical diagnostic criteria for Parkinson's disease. Mov Disord. 2015;30(12):1591-1601. doi:10.1002/mds.26424   Response 1: We thank the reviewer for the suggestions.  We have inserted some additional background information about PD, including new references, but discussion on the role of Orexin and nutritional factors in the pathogenesis of Parkinson's are outside the scope of our review.     Comment 2: The possible significance of DBS in the context of balance should be discussed in the context of subtypes of PD - tremor-dominant/ postural instability    Response 2: The included papers rarely differentiate between the subtypes; hence, it is difficult to make comments about the existing evidence and the role of subtyping in effect of DBS on posturography. Additional comments have been added (Lines 41-44, 84-86, 393-396 on PDF) to discuss the subtypes. Any cases which include descriptions of subtype have also been added to the Supplementary tables.   Comment 3: The information concerning used pharmacological treatment should be discussed more extensively   Response 3: We have discussed the effects of medications on posturography in the discussion (see lines 447-458 on PDF); additional comment has been added to the introduction (see lines 46-49 on PDF). Levodopa equivalent daily dose (LEDD) has also been added to the Supplementary Tables.   Comment 4: The discussed feasibility of DBS should be discussed in the context of other stereotactic methods e.g. Gamma Knife and Focus Ultrasound.   Response 4: We thank the reviewer for this suggestion.  We have added further information to the background (lines 72-74 on PDF), but it is beyond the scope of this paper to analyse the differences between these different interventions on posturography.   Comment 5: The description of methodological limitations is insufficient.   Response 5: As per the reviewer's suggestion, we have expanded on the limitations of this review (added; see lines 517-524); including discussion on heterogeneity being the main limitation and that this should be addressed by introducing an international consensus for posturography. 

Reviewer 2 Report

Comments and Suggestions for Authors

Thank you for giving me the opportunity to review this interesting paper. Postural imbalance with falls affects a large number of patients with PD and therapies are poor at improving this symptom. This study reviewed the quantitative posturography measures and analyses used for assessment of postural imbalance in literature on DBS treatment for PD. The study found that STN- and GPi-DBS reduce static sway in PD and mitigate the increased sway from levodopa. STN-DBS impairs automatic responses to perturbations, whilst GPi- DBS has a more neutral effect.  In conclusion, STN- and GPi-DBS have differing effects on posturography which suggest site-specific and possibly non-dopaminergic mechanisms.

This paper is generally well written and discussed. But considering the different circuit mechanism the authors proposed, a graphical description would help to summarize the authors hypothesis based on current findings.

In line 169, a well-rounded searched should include more data from various databases, though I think it’s very likely that the authors have found most of the related articles, only PUBMED search is not that enough. And a flow chart of literature search should help.

Line 169-170, what do the authors mean by saying “……conducted in April 2023 using the following search terms, without applying filters……”, so the items “Balance DBS”, “Posture DBS” were searched or not? The used MESH terms should be provided.

In lines 179-182, the description of the result is some kind mixed with the methods.

Better provide reference in the table for Table 5 (named static posturography measures) and Tabe 6 (named Dynamic posturography measures).

Comments on the Quality of English Language

Good.

Author Response

Comment 1: This paper is generally well written and discussed. But considering the different circuit mechanism the authors proposed, a graphical description would help to summarize the authors hypothesis based on current findings.   Response 1: Thank you for this helpful suggestion. We have added a new figure (Figure 4) which intends to address this point. In addition, we have reviewed all of our figures and created new Figures 1-3, to help better illustrate our points and to try to improve the paper in general.   Comment 2: In line 169, a well-rounded searched should include more data from various databases, though I think it’s very likely that the authors have found most of the related articles, only PUBMED search is not that enough. And a flow chart of literature search should help.   Response 2: We have performed an updated literature search using multiple databases (Scopus, Web of Science), which has led to the addition of 4 extra papers. This is due to the original search being in April 2023; all new papers have been published in the intervening period. Although adding a flow chart is a helpful suggestion, we have not been able to follow this because of the non-systematic (narrative) nature of our review.   Comment 3: Line 169-170, what do the authors mean by saying “……conducted in April 2023 using the following search terms, without applying filters……”, so the items “Balance DBS”, “Posture DBS” were searched or not? The used MESH terms should be provided.   Response 2: Thank you for this suggestion. Our initial wording was ambiguous- “Deep Brain Stimulation”; “Posture”; and “Postural balance” were the MeSH terms used. We have updated this section to make this clearer.   Comment 4: In lines 179-182, the description of the result is some kind mixed with the methods.   Response 4: Thank you for this feedback. This has now been moved to the correct section within the methods.   Comment 5: Better provide reference in the table for Table 5 (named static posturography measures) and Tabe 6 (named Dynamic posturography measures).   Response 5:  We have added references for most of the individual measures in Tables 5 and Table 6 which are less common and/or more technical. Not every measure has been provided with a reference because some of these measures are used frequently in posturography literature, which are used throughout many papers included in this review.

Reviewer 3 Report

Comments and Suggestions for Authors

This study provides a comprehensive overview of the current literature on the effects of deep brain stimulation (DBS) on postural control in Parkinson’s Disease (PD) patients, with a focus on posturography as a quantitative measure. There are some questions for the authors:

  1. In Figure 2(B), is the plot the “COP”or the force of left or right foot?
  2. In page 5 line 156-157, “Gait initiation APAs can be measured CoP shifts from IMUs and lower limb motor responses from surface electromyography (EMG).”has grammar error, please correct it and also check the manuscript thoroughly.
  3. Why page 6 is nearly blank? Please check it?
  4. “A PUBMED research was conducted in April 2023”, while this is March 2025, therefore, what are the novelties of this manuscript? Should the author added new paper in the latest 2 years?
  5. "Static posturography" and "dynamic posturography," should be defined to be beneficial for readers who may not be familiar with these terms.
  6. In line “361-362”, the authors mentioned that “Only two studies collected data beyond 12 months; thus, we know little about the longterm effects of DBS on postural control and how this compares to disease progression without DBS.” , the corresponding references can be cited here.
  7. The paper could include a more detailed section on future research directions.

Overall, this paper is a valuable contribution to the literature on DBS and postural control in PD. It provides a thorough review of the current evidence and highlights important areas for future research.

Author Response

Comment 1: In Figure 2(B), is the plot the “COP” or the force of left or right foot?   Response 1: We thank the reviewer for pointing this out - this was an error; the image represents CoM data from IMUs. The manuscript has been updated to reflect this.   Comment 2: In page 5 line 156-157, “Gait initiation APAs can be measured CoP shifts from IMUs and lower limb motor responses from surface electromyography (EMG).”has grammar error, please correct it and also check the manuscript thoroughly.   Response 2: We again thank the reviewer for pointing this out - the grammatical error has been corrected.   Comment 3: Why page 6 is nearly blank? Please check it?   Response 3: Page 6 does not appear blank on our Microsoft Word version of the manuscript. Perhaps there was a formatting error when the manuscript was converted to PDF during submission. We have checked this on the re-submission and hope that this appears correctly this time.   Comment 4: “A PUBMED research was conducted in April 2023”, while this is March 2025, therefore, what are the novelties of this manuscript? Should the author added new paper in the latest 2 years?   Response 4: Thank you for this suggestion. We have performed an additional search and have found 4 additional papers that would be included- they have been added to the supplementary tables. The new papers have also been added to the relevant analysis sections.   Comment 5: "Static posturography" and "dynamic posturography," should be defined to be beneficial for readers who may not be familiar with these terms.   Response 5: The relevant sections have been updated as per the reviewer's suggestion with formal definitions, at lines 99-100 and 139-140 on the PDF; thank you for this suggestion.

Round 2

Reviewer 1 Report

Comments and Suggestions for Authors

The pathophysiological aspects were not sufficiently stressed as indicated in the first round.

Comments on the Quality of English Language

Acceptable

Author Response

We thank the reviewer for the comment. We do not feel that extensive discussion of the pathophysiology of Parkinson's, especially with regards to the role of Orexin or nutritional aspect of Parkinson's, is relevant to our manuscript; nor are they relevant to the theme of this special issue, which is "Advances in Deep Brain Stimulation for Parkinson's Disease and Other Movement Disorders".

Reviewer 3 Report

Comments and Suggestions for Authors

All my questions have been addressed well.

Author Response

Thank you for your time and support in improving this work.